# Effects of Energy Drink Consumption on Physical Performance and Potential Danger of Inordinate Usage

**DOI:** 10.3390/nu13082506

**Published:** 2021-07-22

**Authors:** Jakub Erdmann, Michał Wiciński, Eryk Wódkiewicz, Magdalena Nowaczewska, Maciej Słupski, Stephan Walter Otto, Karol Kubiak, Elżbieta Huk-Wieliczuk, Bartosz Malinowski

**Affiliations:** 1Department of Pharmacology and Therapeutics, Faculty of Medicine, Collegium Medicum in Bydgoszcz, Nicolaus Copernicus University, M. Curie 9, 85-090 Bydgoszcz, Poland; wicinski4@wp.pl (M.W.); eryk.wodkiewicz09@gmail.com (E.W.); bartosz.malinowski@cm.umk.pl (B.M.); 2Department of Otolaryngology, Head and Neck Surgery, and Laryngological Oncology, Faculty of Medicine, Collegium Medicum in Bydgoszcz, Nicolaus Copernicus University, M. Curie 9, 85-090 Bydgoszcz, Poland; magy_mat@by.onet.pl; 3Department of Hepatobiliary and General Surgery, Faculty of Medicine, Collegium Medicum in Bydgoszcz, Nicolaus Copernicus University, M. Curie 9, 85-090 Bydgoszcz, Poland; maciej.slupski@cm.umk.pl; 4Department of Urology, Raphaelsklinik, 48143 Münster, Germany; stephotto@gmx.de; 5Department of Obstetrics and Gynecology, St. Franziskus-Hospital, 48145 Münster, Germany; karolkubiak85@googlemail.com; 6Department of Health Promotion, Faculty of Physical Education and Health, Józef Piłsudski University of Physical Education, 00-809 Warsaw, Poland; elzbieta.wieliczuk@awf.edu.pl

**Keywords:** energy drink, taurine, caffeine, physical performance, athlete, side effects, prevalence

## Abstract

The rise in energy drink (ED) intake in the general population and athletes has been achieved with smart and effective marketing strategies. There is a robust base of evidence showing that adolescents are the main consumers of EDs. The prevalence of ED usage in this group ranges from 52% to 68%, whilst in adults is estimated at 32%. The compositions of EDs vary widely. Caffeine content can range from 75 to 240 mg, whereas the average taurine quantity is 342.28 mg/100 mL. Unfortunately, exact amounts of the other ED elements are often not disclosed by manufacturers. Caffeine and taurine in doses 3–6 mg/kg and 1–6 g, respectively, appear to be the main ergogenic elements. However, additive or synergic properties between them seem to be implausible. Because of non-unified protocol design, presented studies show inconsistency between ED ingestion and improved physical performance. Potential side effects caused by abusive consumption or missed contraindications are the aspects that are the most often overlooked by consumers and not fully elucidated by ED producers. In this review, the authors aimed to present the latest scientific information on ED components and their possible impact on improving physical performance as well as to bring emphasis to the danger of inordinate consumption.

## 1. Introduction

Energy drinks (EDs) are widely available beverages containing caffeine, taurine, glucose, and other ingredients and are said to improve physical or cognitive performance and should not be confused with isotonic drinks or sport drinks [1]. EDs first appeared in Europe and Asia in the 1960s. An exponential growth in sales was observed after the introduction of Red Bull to the European market in 1987, which was also observed 10 years later in USA. EDs have become the fastest growing product in beverage category since the introduction of bottled water, and Red Bull is still one of the world’s leading brands [2,3].

In 2013, a study conducted in 16 European countries (*n* = 52,016) recorded a 68% lifetime prevalence of ED use among adolescents (aged 10–18 years old). Among the declared users, 12% were “high chronic” consumers with an average consumption of 7 litres per month. In adults, the prevalence of consumption was 30%, and it 18% in children (aged 6–10 years old) [3]. In 2014, similar results were observed in Canada (*n* = 8210), where 62% of adolescents reported consuming EDs at least once in the previous year [4]. Likewise, in 2019, Galimov and colleagues [5] (*n* = 6902) reported that 61.7% of German adolescents between 9–19 years of age had used ED at least once. Degirmenci et al. [6] noted that 52.3% of Norwegian adolescents (aged 12–19 years old) drank at least one ED per month (*n* = 31,081). Presented data show that a young population, mainly between 9–19 years old, constitutes the biggest pool of ED consumers, with the prevalence of lifetime use ranged from 52% to 68%.

The prevalence of ED consumption among athletes, especially professional sportspersons, is limited and varies across studies. Cross-sectional analysis of survey data of a cohort-study of 2287 young adults (mean age 25.3 years) demonstrated that men who spend more time engaged in physical activity had higher ED consumption and were more likely to use muscle-enhancing substances [7]. In their study in elite young UK athletes (*n* = 403, mean age 17.66 ± 1.99), Petróczi et al. [8] showed that 41.7% of the examined population declared a single ED use; thus, it was the most often chosen supplement among this group. Nowak et al. [9] reported that 69% of Polish students attending sport classes (*n* = 707, mean age 14.3) consumed EDs. Most often, they drank 250 mL daily. Moreover, the same authors noted that adolescents who practiced sports were more willing to drink EDs, and 28% of them claimed that EDs gave them a boost of energy [10]. On the other hand, a study from Gallucci et al. [11] in a US student population found no difference in ED consumption comparing 205 student athletes vs. 487 non-athletes (mean age = 20.4). There was no difference in ED consumption based on athlete status. Heavy episodic drinking and prescription stimulant misuse were both correlated with increased ED consumption. The lack of significance based on sport participation could be due to a low prevalence of heavy episodic drinking among athletes in the sample (14.4% among athletes, 23.3% among non-athletes). Previous studies have also linked ED consumption with heavy episodic drinking among student non-athletes [12] and student athletes [13], supporting the hypothesis.

The high prevalence of ED consumption among the young population may result from a well-conceived marketing strategy that persuades them to purchase and consume EDs [14]. One of those marketing tools is sponsorship. Numerous ED brands are known for offering financial help to many sport disciplines, especially those described as “extreme”. These activities are perceived as attractive [15] and often involve speed, agility, and high levels of physical exertion, which are of interest to young adults. Supporting traditional sports is a well-known practice, but there is a rising trend of providing sponsorships to eSport teams [16,17]. The popularity of tournaments that are considered to be eSports is drastically raising in the abovementioned age groups [18]. In 2017, Twitch, a live streaming platform for gamers, had a higher viewership when compared to CNN and MSNBC, and had a viewership that was to similar to ESPN at any given time [19]. ED consumption has become increasingly popular among eSports contestants, but the detailed prevalence was not specified [20]. The aim of ED consumption in this context is to sustain peak cognitive performance by professional video gamers.

As EDs have risen in popularity over the years, particularly among young people, considerable debate should take place over the validity and safety of their intake. This review is an attempt to describe the present state of knowledge about the ED components and their biochemical and ergogenic properties as well as to emphasize to the danger of the short- and long-term adverse effects of inordinate consumption.

## 2. ED Ingredients—Caffeine and Taurine Contribution

ED compositions vary widely, and manufacturers often do not disclose the exact quantities of particular ingredients. Furthermore, ED companies do not distinguish which component, alone or in combination, explains the potential positive benefits of the drink [2,3]. However, it appears to be obvious that EDs mainly contain caffeine, which may be combined with other ingredients such as sweeteners (natural or artificial), glucuronolactone, amino acids (mostly taurine, sometimes L-carnitine), B-vitamins, and herbal supplements such as ginseng or biloba [1,9]. Additional amounts of caffeine can be present in other compounds declared on ED labels such as guarana, yerba mate, or kola nuts [9]. On average, the caffeine content in EDs ranges from 75 to 240 mg compared to the 77–150 mg of caffeine in a cup of coffee [10,21].

The ergogenic effects of caffeine are connected to its biological activity in the central nervous system and in the peripheral organs. Being a nonselective adenosine receptor antagonist, caffeine acts as central nervous system stimulant via the inhibition of the sleep-promoting adenosine effect that results in the release of dopamine, serotonin, and norepinephrine in the brain [22]. The secretion of these neurotransmitters may cause improved mood, alertness, fatigue delay, and psychomotor coordination [22,23]. Moreover, caffeine via the adenosine receptors subtypes A_1_R and A_2a_R in the brain can modulate proinflammatory cytokine production. This property might interfere with the inflammation process after physical exercises, when the levels of cytokines, mainly TNF-α, IL-1β, IL-6, and IL-10, are elevated [23]. The effect of caffeine on peripheral tissues is more comprehensive. In muscle cells, caffeine and its metabolites have the ability to enhance Ca^2+^ release into the intracellular space. This effect is possibly achieved by caffeine acting as an antagonist of the adenosine receptor subtype A_1_ on the skeletal muscle membrane or/and attachment to ryanodine receptors (RyRs) on the sarcoplasmic reticulum (SR) [23,24]. Besides the greater release of Ca^2+^, increased myofibrillar Ca^2+^ sensitivity, improved SR Ca^2+^ permeability, and decreased activity of the SR Ca^2^ pump are attained, ultimately resulting in modified muscle performance [24,25]. Furthermore, studies have shown that caffeine administration leads to an increase in circulating epinephrine, norepinephrine, cortisol, plasma lactate, whereas the plasma beta-endorphin can increase by almost two-fold [26]. These effects may impact exercise endurance by lowering pain perception and affecting the use of the energy substrates-promotion of free fatty oxidation, glycogen storage, and decreased insulin sensitivity [26,27]. However, these metabolic responses and their impact on the muscles remains unclear. It is well documented that a dose of caffeine of 40 to 60 mg improves cognitive functions, whereas a minimal dosage of 3 mg/kg is required to enhance physical performance [28]. Systematic reviews and meta-analyses highlight an ergogenic effect of caffeine ingestions (3–6 mg/kg) in such exercise tests as: 1 repetition maximum (1 RM) strength, isokinetic peak torque, vertical jump height, power output, aerobic and muscular endurance [29]. In their meta-analysis, Salinero et al. [30] demonstrated that the use of caffeine on team sports improved total running distance, distance covered at sprint velocity, and the number of sprints. Furthermore, in their systemic review, Mielgo-Ayuso et al. [31] showed no differences between the sexes in aerobic performance and the fatigue index after caffeine administration. However, more power during lifting and sprinting was produced in male athletes than in female athletes. Beyond the physical aspect, it was found that a low or moderate dose of caffeine can improve self-reported energy, mood, and cognitive functions such as attention in sports practice [32]. These reports show that caffeine administration may be an important factor in competitive performance at an elite level where results depend on narrow margins and may also be helpful for amateurs in beating their personal records. In summary, many of the proven caffeine-related effects are consistent with those stated by ED producers.

The next key component of ED is taurine, a sulphur-containing amino acid widely distributed in tissues and naturally produced by the human body [33]. Dietary supplements contain a synthetic version of taurine, whilst in meat, seafood, and fish it occurs naturally. Plants are not abundant in taurine, thus in vegans, its tissue and circulating levels are about 20% lower than those of meat-eaters [34]. The mean daily intake of taurine in adults is estimated to be between 40 and 400 mg [35]. As the quantity of taurine is often not disclosed on the ingredient list, a study based on 10 brands of EDs presented taurine content with mean value 342.28 mg/100 mL [36]. These results suggest that the intake of taurine from EDs is several times higher than that from other types of food.

The physiological activity of taurine is widespread and includes the modulation of mitochondria function, bile acid conjugation, the regulation of cellular calcium levels, and cytoprotective properties [37]. This amino acid is found in high concentrations in the brain and in the skeletal muscles, where it plays an important role in neurotransmission and in modulation of contractile function [33,38]. Taurine associated actions also appear to be connected to several hormones, including norepinephrine, dopamine, and growth hormone, but the mechanism of direct influence on the secretion of those hormones remains unclear [39]. Studies show that there are a few possible biochemical mechanisms of taurine in modulating muscle contractile function. First, using type I and type II human fibers, Dutka et al. [38] presented that intracellular taurine modified sarco(endo)plasmic reticulum Ca^2+^-ATPase (SERCA) function, either via an augmentation in its maximum SR Ca^2+^ pumping rate or its affinity for Ca^2+^, which finally led to an increase in the rate of Ca^2+^ accumulation by the SR. However, the maximum Ca^2+^ content produced by the SR was not affected. SR Ca^2+^ accumulation was not changed when the intracellular taurine concentration was higher than physiological range, but lowering the taurine level below its normal range resulted in significantly decreased SR Ca^2+^ content. This observation led authors to speculate that such a decrease in the SR Ca^2+^ level would adversely affect muscle function because of its slower relaxation. Second, intensive activity of the skeletal muscle cells leads to an increased production of reactive oxygen species (ROS), which contribute to contractile proteins oxidation, including the inhibition of SERCA [40]. Taurine appears to prevent ROS generation at the mitochondrial respiratory complex I level [41]. This process is possibly achieved through the modification of the mitochondrial tRNA caused by taurine, which affects the synthesis of proteins engaged in effective electron transfer in complex I [42,43]. Besides proteins oxidation, ROS in combination with the hypoxic state in overused muscles induce DNA damage, which may be prevented by taurine in NF-kB signaling pathway or regulating Ca^2+^ homeostasis [41]. The taurine seems to have a role in multiple intracellular activities, and these exact molecular mechanisms need to be investigated.

Whilst the impact of caffeine on physical performance is well-established, the evidence of taurine efficacy is limited. A systemic review based on 10 studies with 128 volunteers, showed that an isolated dose of taurine in varying amounts (1–6 g), regardless of acute or chronic supplementation, has the potential to improve overall endurance performance (Hedges’ g = 0.4; *p* = 0.004). Moreover, the taurine dose did not have an impact on its ergogenic effect and could be as low as 1 g or as high as 6 g. The differences between sexes were not investigated [44].

It is logical to hypothesize that a combination of caffeine and taurine may possess additive or synergic properties. The mentioned meta-analyses allow the assumption that an isolated caffeine dosage of 3–6 mg/kg [28,29,30] or a taurine intake of 1–6 g [44] (which is in consonance with ~14–86 mg/kg if the presumed body mass is 70 kg) are enough to trigger an ergogenic effect in human beings. If an additive or synergic effect exists between these two supplements, one of two changes should be observed: The co-administration of lower doses would enhance physical performance, or the attained improvement would be equal to or greater than the sum of the effect of the two substances taken separately in the same quantity.

A study by Tallis et al. [45] was conducted using an in vitro model using isolated mouse soleus muscle. The ergogenic effects of the physiological concentration of taurine, caffeine, and the combination of caffeine and taurine were compared. Exposure to taurine did not provide any acute ergogenic effect, whereas treatment with caffeine alone and in combination with taurine showed a significant increase in acute muscle power output and decreased time to fatigue. What is important is that the increase achieved by the combination did not exceed the sole caffeine impact. Likewise, Chaban et al. [46] applied an in vitro model based on human myocardial tissue to evaluate the positive inotropic effect of taurine, caffeine, or both when administrated in the concentration levels corresponding to a high but non-toxic plasma level. The results of the study also presented that taurine did not alter the contraction, whilst a significant increase was gained by caffeine alone and in combination with taurine. However, exposure to caffeine did provide similar results to those achieved by the combination of both.

A study of Warnock et al. [47] showed that isolated caffeine (5 mg/kg), taurine (50 mg/kg), and a combination of caffeine and taurine (5 mg/kg and 50 mg/kg, respectively) improved performance during repeated Wingate tasks (repeat-sprint cycling) compared to the placebo, by increasing mean power, peak power, and mean peak power. Every participant made four visits separated by 48 h, performing three 30-s Wingates divided by 2-min of active recovery during each visit. Interestingly, taurine administration led to slightly better results than the use of caffeine alone, or even with a combination of both, but the presented differences were still small. However, the single-blind research design and the group of seven male team sport players seemed to be limiting [47].

On the other hand, in their double blind, cross-over study (*n* = 11), Jeffries et al. [48] achieved no significant differences between administered supplements and a placebo in male participants who completed sprints on a cycle ergometer after ingesting a caffeine (80 mg)/taurine (1 g) combination or a placebo. Based on the other abovementioned research, taurine and caffeine doses of 11–15 mg/kg and 0.8–1.2 mg/kg, respectively, seem to not be enough to trigger any ergogenic effect. What is more, the study did not include measurements after the administration of the compounds separately. 

Similar results were obtained by Kammerer et al. [49] in 14 male volunteer soldiers who consumed either placebo, caffeine (80), taurine (1000 mg), a combination of caffeine and taurine (80 mg + 1000 mg), and a commercial drink with the same amounts of caffeine and taurine. None of the interventions improved performance in physical tests such as maximum oxygen consumption (VO2max), time to exhaustion, isometric strength, and vertical jump. As in the research of Jeffries et al. [48], taurine and caffeine doses (~11–15 mg/kg and 1–2 mg/kg, respectively) could be insufficient. 

A study by Aggarwal et al. [50] in 18 sleep-deprived medical students evaluated the impact of taurine and caffeine administration on their psychomotor skills with the use of simulated laparoscopy (Minimally Invasive Surgical Trainer). Subjects who consumed caffeine (150 mg) or a combination of caffeine and taurine (2 g) demonstrated shorter reaction times and higher motion economy compared to the placebo group, but the number of errors was not reduced. Unfortunately, doses of supplements were not presented in mg/kg of body mass, and the ingredients were not ingested separately.

The presented investigations suggest that the contribution of two primary ED ingredients possibly do not provide any additive or synergic effect. This lack of reaction may be connected to the potential molecular or hormone path shared between both caffeine and taurine [26,39]. However, more studies are required to check if this kind of combination triggers any effect. In addition, conducted analysis should include various doses of caffeine and taurine ingested separately and in combination compared to a placebo.

The effects of other ED ingredients such as glucuronolactone, carnitine, guarana, ginseng, and vitamins on physical performance have not been tested alone on humans, or the current experimental evidence is inconsistent and limited [28]. Only yerba mate, which contains caffeine (17.5 mg/gram) and flavonoids, seems to help with reducing weight and increasing fatty acid oxidation during exercise [1]. Carbohydrate concentration in EDs (11–12%) appears to be ineffective for quick absorption and hydration during physical activity. Optimal sugar content for athletes constitutes 6–8% and is found in sport drinks. This specific amount of carbohydrates make the water absorption in intestine more efficient and may be an exogenous source of carbohydrate oxidation [28,51]. 

## 3. Energy Drinks in Sport Performance

Due to the popularity of energy drinks and their potential ergogenic effect, many studies undertake the topic of correlation between ED ingestion and improved physical performance. In their meta-analysis of 34 studies published between 1998 and 2015 with 653 subjects, Souza et al. [52] demonstrated that ED ingestion improved physical performance in the following activities: muscle strength and endurance (effect sizes (ES) = 0.49; *p* < 0001), endurance exercise tests (ES = 0.53; *p* < 0.001), jumping (ES = 0.29; *p* = 0.01), and sport-specific actions (ES = 0.51; *p* < 0.001), but it did not any improved physical performance in sprinting (ES = 0.14); *p* = 0.06). Interestingly, a significant association between taurine dosage and increased physical performance (*p* = 0.04) was found, but not between physical performance and caffeine dosage (*p* = 0.21). Unfortunately, the EDs contained other potentially ergogenic elements, which may have influenced the final results, but were not included in the comparison.

Prins et al. [53] conducted a double-blind, crossover study on 18 recreational runners who were randomized to supplement with 500 mL Red Bull (2 cans) and sugar-free placebo 60 min before completing a 5-km time trial on a treadmill separated by 7 days. The study resulted in 2.12% (~30 s) improvement in the ED group compared to the placebo group (*p* = 0.016). A total of 78% of the participants performed better after ED ingestion. The amount of caffeine ingested was in the range of 1.5–3.9 mg/kg (average 2.9 mg/kg), and the relation between caffeine intake and the 5-km running time demonstrated a weak correlation. The average amount of consumed taurine was 28 mg/kg, but the association between its dose and physical outcomes was not investigated.

A double-blind study by Collier was performed on 15 physically active students who were familiar with resistance training. Every participant served as their own controls and were tested across three trials, each separated by 7 days, ingesting either a caffeinated, uncaffeinated, or a placebo beverage. The amount of caffeine received from the caffeinated ED was 5 mg/kg, whereas the exact taurine dose was not disclosed. The results of the study presented a 5% improvement in the maximum voluntary isometric contraction strength of the knee extensors after the ingestion of a caffeinated ED compared to the placebo (*p* = 0.015). This effect was maintained even after an exhaustive bout of high-force contractions. However, the strength change for the uncaffeinated ED was not significantly different from the caffeinated ED, suggesting that caffeine was not the sole ergogenic agent in the used beverages [54]. 

Research performed by Jacobson et al. [55] on 36 physically active undergraduate students showed that the stimulant dosage in EDs was enough to enhance the performance of smaller muscle groups (forehand stroke; *p* < 0.05) but was not sufficient to affect the larger muscle groups of the lower legs (countermovement vertical jump; *p* > 0.05). Moreover, a comparison of gender differences implied that only women improved their forehand stroke velocity after ED intake, while the results of a vertical jump test were similar between the sexes. The amount of caffeine in the EDs was constant and amounted about 240 mg, and the men were over 23% heavier than the women. Thus, the men received an average dose of 2.8 mg/kg, whereas the women ingested an average dose of nearly 4.0 mg/kg. The taurine dose was not disclosed, but other ingredients were also present. Presumably, the variabilities in body mass, body composition between sexes, and caffeine distribution in various tissues in proportion to their water content could cause the abovementioned differences.

Gallo-Salazar et al. [56] conducted a double-blind study on 14 elite junior tennis players who ingested either a caffeinated ED or an uncaffeinated ED (placebo) in two different sessions separated by 1 week. The amount of caffeine was 3 mg/kg in the caffeinated ED, whereas the taurine dose was 18.7 mg/kg in both drinks. Both EDs contained other potentially ergogenic ingredients. In comparison to the placebo beverage, the caffeinated ED increased handgrip force by 4.2% ± 7.2% (*p* = 0.03) in both hands, running pace at high intensity (*p* = 0.02), and the number of sprints (*p* = 0.05) during the simulated match. There were tendencies for increased maximal and mean running velocity tests and higher percentage of points won on service with the caffeinated ED, although the differences did not reach statistical significance. However, during the serving test, no improvements were found in ball velocity and total distance. The limitation of the study was lack the of placebo beverage without potentially active components.

On the other hand, a number of published studies failed to report statistically significant ED influence on physical performance. A double-blind, crossover study of Fernández-Campos et al. [57] conducted on 19 professional female volleyball players measured grip strength, vertical jump, and anaerobic power in three different sessions (ED, placebo, or no beverage). The approximate amount of ingested caffeine was 2 mg/kg, whereas the approximate taurine dose was 2 g (~30 mg/kg). The results showed no significant improvement for each performance test, except in right hand grip strength, which was significantly higher in the ED condition compared to the placebo and no beverage conditions (*p* = 0.25, *p* = 0.006, respectively). However, the authors concluded that the acute ingestion of the ED did not improve the physical performance of the female volleyball players. It is worth mentioning this was the second study that presented that smaller muscle groups tend to be more sensitive to ED ingredients than larger muscles [55].

Al-Fares et al. [58] conducted a single-blinded, crossover study on 32 healthy untrained female students who consumed either an ED or a placebo in two trials. During each session, time to exhaustion and VO2max were measured on a treadmill. Each participant ingested an ED with equal doses of caffeine (160 mg, ~3.1 mg/kg), taurine (2 g, ~38.7 mg/kg), and other active ingredients. The results of the study showed no significant differences in time to exhaustion (*p* < 0.157) and VO2max (*p* < 0.154) compared to the placebo. However, this study presented the effects of EDs on an untrained population in comparison to amateurishly or professionally trained participants in previous studies. Moreover, a limitation of the study was single-blinded design.

Presented investigations show that EDs have the potential to improve physical performance. However, the mechanism of this enhancement and main substance responsible for that effect maintains unclear. A meta-analysis of 34 studies highlighted a significant association between taurine dosage and better performance [52]. A similar hypothesis was stated by Collier et al. [54], stating that taurine was supposedly the main stimulant. In contrast, the study by Prins et al. [53] found a weak correlation between caffeine and improved physical performance, but taurine association was not investigated. In addition, the study on tennis players presented better results in a group where a caffeinated ED was ingested compared to an uncaffeinated ED [56]. These contradictions may arise from the difficulties of conducting repetitive studies. Similar inconsistent observations appeared in the investigations, which examined the interaction between caffeine and taurine [47,50]. Possible reasons for discrepancies are the following: (1) the variability of ED ingredients and their exact doses, which are often not disclosed by manufacturers; (2) a high number of bioactive components in EDs and interactions that may exist between them, (3) the classification of participants as mild, moderate, heavy ED users or non-users, which may impact their sensitivity to ED ingredients; (4) weight-based dosing; (5) the non-identical assessment of physical skills, endurance and strength; (6) a wide spectrum of examined sport disciplines; and (7) variability in the protocol design.

Summarizing the results of this part, adult athletes without comorbidities who are willing to improve their physical performance may consider using isolated caffeine in a minimal 3 mg/kg dosage ingested 30–60 min before exercise [28,29,30,31] or sport drinks (hypotonic drink before training and isotonic drink during exercise) [28,51]. The conducted investigations presented that these supplements have legitimate athletic value and are well-tolerated, thus with appropriate guidance, they can be used by sportsmen. As long as data do not fully support positive ED impact on physical performance, they should be considered as a secondary choice with awareness of their side effects.

The summary of the studies describing the impacts of EDs on physical performance is shown in Table 1.

## 4. Alarming Effects Associated with Energy Drink Consumption

The popularity of EDs, especially among adolescents and young adults, has brought many questions and concerns about their overall safety and health in the scientific community. Tempting slogans and sponsorship in extreme sports and e-Sports are cleverly used by ED campaigns to promote their products without the necessity of mentioning their side effects [14,19,20]. Unwanted reactions can range from mild to life-threatening conditions, and these vary from person to person. EDs are often consumed during conditions with increased circulating catecholamines, such as during sport performance, sleep deprivation, and stress-related situations, which may potentiate the stimulant effect and lead to more serious adverse effects. In addition, adolescents may be more sensitive to caffeine and other stimulant components since their body weight is much lower [3,4,5,6].

The short- and long-term effects of the ED consumption are arguable, although the short-term effects appear to be dominant. The most common adverse effects are associated with the cardiovascular and neurological systems [59]. A meta-analysis of 96,549 participants demonstrated that the most frequent reported adverse events among pediatric patients were insomnia (35.4%), stress (35.4%), and depressive mood (23.1%), whilst among adult population, the most reported adverse effects were restlessness/jitteriness/shaking hands (29.8%), insomnia (24.7%), and gastrointestinal upset (21.6%). For both groups, the most common side effects for the particular systems were tachycardia for the cardiorespiratory system, headaches for the neurological system, insomnia/sleeping-related symptoms for the psychological system, and increased urination for the renal system [60]. Since mild adverse effects may be acceptable to the ED users, severe complications are unwanted.

According to various reports, EDs can cause all kinds of arrhythmias, with occurring atrial fibrillation (AF) the most often [61]. QT prolongation, aortic dissection, myocardial infarctions, acute coronary vasospasm, and sudden death have been also observed in healthy subjects with structurally normal hearts or high-risk cardiovascular patients due to massive quantities of EDs consumption [62,63,64]. As a long-term effect, EDs are able to induce morphological changes in the heart muscle. Similar to those related to ethanol consumption, which may lead to disrupted cardiac activity [65].

A significant connection between ED consumption and mental health problems among adolescents has been found in several studies. They were associated with a greater risk of moderate and serious levels of psychological distress, suicidal thoughts, and suicide attempts [66]. Similar findings were found in Korean adolescents who suffered from depression and suicide ideation, which were related to excessive ED consumption [67]. Although women are more vulnerable to mental health problems, studies presented that this association was stronger in males than females, possibly due to higher ED consumption among young men [66,67]. Moreover, some scientist started to recommend considering screening for ED use in assessments of mental health, especially for young men with diagnosed anxiety and stress symptoms [68].

The reported metabolic impact of EDs is contradictory. A recent prospective study with a two-year follow-up duration (measurements were taken at 20-years and at 22-years) did not find compelling evidence between frequency of ED-intake, BMI, metabolic syndrome, blood pressure, fasting glucose, triglycerides, and HDL [69]. However, the presence of young subjects and the two-year period of observation seem to be inadequate to develop such metabolic disorders. On the contrary, Shearer et al. [70] presented that caffeinated beverages could cause acute, transient insulin resistance in healthy adolescents. Moreover, ED consumers were also more likely to consume highly energy-dense fast foods and spend more than 6 h in front of a screen, which may result in an increased risk of being overweight or obese among adolescents [71]. Likewise, the next study showed that the regular consumption of EDs could contribute to the development of obesity and dental erosion because of the high sugar content and ED pH below 5.5 [72].

In addition, a recent study on 129,809 Korean adolescents presented that frequent ED consumption was associated with allergic diseases, such as asthma, allergic rhinitis, and atopic dermatitis [73]. The last finding also demonstrated that EDs containing caffeine can stimulate gastrin and gastric acid secretion and may reduce the competence of the lower esophageal sphincter, which eventually leads to gastroesophageal reflux disease (GERD) [64,73]. Interestingly, a case study revealed that atrophic gastritis and the gastric intestinal metaplasia can be limited with the withdrawal of EDs [74].

Beyond the previously mentioned alarming effects of ED consumption, there are extra concerns for athletes. EDs induce diuresis and slow fluid absorption, which may exert a negative influence in particularly during hot weather and during ultra-resistance sports competitions like marathons [28,51,64]. In addition, dehydration was pointed out as a patomechanism of exercise-induced asthma because a reduced airway surface protects against further liquid loss [73]. As shaking hands is the one of the most common side effects [60], it may exert a negative effect on the final outcome in sport disciplines where precision is required. On the other hand, a study where an association between ED consumption and mental problems was investigated presented that young people who ingested EDs and were engaged in physical activity reported less symptoms of depression [75]. Similar findings were observed in previously mentioned studies where none of subjects experienced any mental side effect [53,57,58].

Unclear pharmacology and interactions between various ingredients found in these drinks put children at risk of dangerous consequences [64]. Considering that adolescence is an important stage of physical and psychological development, health care providers and teaching staff should inform both parents and adolescents that EDs have no therapeutic benefit at a young age and may lead to serious health events, including sudden death. This prudence also involves young athletes, who may look for an easy boost to augment their physical performance. Manufacturers should be transparent and should inform consumers that their products are associated with harmful short-, long-term effects and warn against intentional overuse. In summary, EDs are not recommended for children and adolescents

The summary of the studies describing the adverse effects of energy drink consumption is shown in Table 2.

## 5. Conclusions

According to this review, high-quality studies are undeniably needed. One of the main aspects that should be taken into account is evidence suggesting a potential interaction between caffeine and taurine. This evaluation would answer the question whether using these supplements together is beneficial and would mark a direction for subsequent studies of ED ingredients. Moreover, knowledge of these interactions would make side effects more predictable in case of inordinate usage. Due to different protocol design, there are many incompatible studies about the correlation between ED usage and exercise performance in athletes. It is still uncertain if ED supplementation should be recommended and what doses of particular ingredients should be added to trigger optimal improvement among athletes. Only the unification of protocol design will significantly answer the question of whether EDs can effectively help in improving physical performance.

## Figures and Tables

**Table 1 nutrients-13-02506-t001:** Energy drink and sport performance.

Author	Research Design	Study Group	Energy Drink	Caffeine Doses	Taurine Doses	Time of Ingestion before the Experimental Session	Performance Metric	Results
Prins et al. 2016 [53]	Double-blind, crossover	*N* = 18 (13 M, 5 W) Recreational runners; 20.39 ± 3.27 years; 71.25 ± 17.17 kg; 178.00 ± 7.57 cm	2 × Red Bull	~2.9 mg/kg 160 mg	2 g	60 min	5-km running	Significant difference ED 1413.2 ± 169.7 vs. PL 1443.6 ± 179.2 s; *p* = 0.016
Collier et al. 2016 [54]	Double-blind	*N* = 15 (7 M, 8 W) Physically active students; 26.1 ± 3.5 years; 70.7 ± 12.1 kg; 174.6 ± 6 cm	Full Throttle	5 mg/kg	Present, not disclosed	30 min	Maximum voluntary isometric contraction strength of the knee extensors	Significant difference ED 5.0 ± 1.7 vs. PL 0.5 ± 1%; *p* = 0.015
Jacobson et al. 2018 [55]	Double-blind, independent groups	*N* = 36 (17 M, 19 W) Physically active students; age range = 19–26 years	5 h Energy	240 mg	Present, not disclosed	30 min	Peak velocity of isolated forehand stroke	Significant difference ED +5.9 vs. PL −1.9%; *p* < 0.05
							Average velocity of isolated forehand stroke	Significant difference ED +8.6 vs. PL −1.1%; *p* < 0.05
							Peak power of countermovement vertical jump	No significant difference ED +2.3 vs. PL −0.2%; *p* > 0.05
							Peak velocity of countermovement vertical	No significant difference ED +2.9 vs. PL −0.1%; *p* > 0.03
Gallo-Salazar et al. 2015 [56]	Double-blind	*N* = 18 (13 M, 5 W) Recreational runners; 20.39 ± 3.27 years; 65.2 ± 10.6 kg; 174.4 ± 9.5 cm	Fure	3 mk/kg	18.7 mg/kg	60 min	Right handgrip force	Significant difference ED 402 ± 83 vs. PL 387 ± 83 N; *p* = 0.03
							Left handgrip force	Significant difference ED 361 ± 74 vs. PL 348 ± 76 N; *p* = 0.03
							Maximal serve velocity	No significant difference ED 42.7 ± 50 vs. PL 42.6 ± 4.8 m/s; *p* = 0.49
							Mean serve velocity	No significant difference ED 41.4 ± 5.2 vs. PL 41.6 ± 5.1 m/s; *p* = 0.41
							Maximal running speed	No significant difference ED 22.9 ± 2.1 vs. PL 22.3 ± 2.0 km/h; *p* = 0.07
							Mean running speed	No significant difference ED 21.3 ± 1.5 vs. PL 20.7 ± 2.2 km/h; *p* = 0.12
							Running pace at high intensity	Significant difference ED 63.3 ± 27.7 vs. PL 46.7 ± 28.5 m/h; *p* = 0.02
							Distance	No significant difference ED 2904 ± 430 vs. PL 3058 ± 620 m/h; *p* = 0.24
							Sprints	Significant difference ED 13.2 ± 1.7 vs. PL 12.1 ± 1.7 number/h; *p* = 0.05
							Peak running velocity	No significant difference ED 20.5 ± 2.8 vs. PL 19.5 ± 2.3; *p* = 0.44
Fernández-Campos et al. 2015 [57]	Double-blind, crossover	*N* = 19 (19 W) Professional volleyball players; 22.3 ± 4.9 years; 65.2 ± 10.1 kg; 171.8 ± 9.4 cm	Unknown	2 mg/kg	2 g	30 min	Right hand grip strength	Significant difference ED vs. PL *p* = 0.025 ED vs. no beverage *p* = 0.0025
							Left hand grip strength	No significant difference
							Countermovement jump	No significant difference
							Squat jump	No significant difference
Al-Fares et al. 2015 [58]	Single-blind crossover	*N* = 32 (32 W) Untrained students; 19.9 ± 0.8 years; 51.7 ± 3.7 kg; 156.4 ± 3.8 cm	Unknown	~3.1 mg/kg 160 mg	38.7 mg/kg2 g	45 min	Time to exhaustion	No significant difference ED 11.41 ± 1.56 vs. PL 11.67 ± 1.51 min; *p* < 0.157
							Maximum oxygen consumption (VO2max)	No significant difference ED 34.06 ± 6.62 vs. PL 32.89 ± 6.83 min; *p* < 0.154

M-male; F-female; ED- energy drink; PL-placebo.

**Table 2 nutrients-13-02506-t002:** Energy drink consumption and associated adverse effects.

Author	Research Design	Study Group	Adverse Effects
Ali et al. 2015 [59]	Systemic review	*N* = 43 reports	The most common associated with cardiovascular (arrhythmias) and neurological (seizures) systems
Nadeem et al. 2021 [60]	Systemic review and meta-analysis	*N* = 96,549 participants	Pediatric population: insomnia (35.4%), stress (35.4%), depressive mood (23.1%) Adult population: restlessness/jitteriness/shaking hands (29.8%), insomnia (24.7%), gastrointestinal upset (21.6%)
Özde et al. 2020 [61]	Prospective, observational, open-label study	*N* = 54 young adults	Prolonged atrial electromechanical conduction times: atrial electromechanical coupling-lateral, atrial electromechanical coupling-septal, intra-atrial electromechanical delay
Ullah et al. 2018 [62]	Case report	Healthy 25-year-old man	Myocardial infarction
Masengo et al. 2020 [66]	Cross-sectional study	*N* = 5538 middle and high school students	Association with greater risk of moderate and serious levels of psychological distress, suicidal thoughts, suicide attempts
Kim et al. 2020 [67]	Cross-sectional study	*N* = 53,312 adolescents	Association with depressive mood and suicide ideation
Kaur et al. 2020 [68]	Longitudinal cohort study	*N* = 429 young adults	After 2-year follow-up, males had increase in depression, anxiety, and stress scores
Trapp et al. 2020 [69]	Prospective study	*N* = 1117 young adults	After 2-year follow-up, no significant associations between energy drink consumption and BMI or metabolic syndrome
Shearer et al. 2020 [70]	Double-blind, randomized cross-over study	*N* = 20 adolescents	Acute, transient insulin resistance
Almulla et al. 2020 [71]	Cross-sectional study	*N* = 1611 school students	Association with reduced sleep duration and increased fast foods consumption
Wee et al. 2020 [73]	Cross-sectional study	*N* = 129,809 adolescents	Association with allergic diseases such as asthma, allergic rhinitis, and atopic dermatitis
Garg et al. 2020 [74]	Case report	35-year-old woman	Energy drinks as a possible risk factor of atrophic gastritis and gastric intestinal metaplasia
Tóth et al. 2020 [75]	Cross-sectional study	*N* = 642 high school and college students	The most common effects: tachycardia, insomnia, tremors, headache

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
