# Peer review of "Effects of Energy Drink Consumption on Physical Performance and Potential Danger of Inordinate Usage"

_nutrients, 2021, doi:10.3390/nu13082506_

Round 1

Reviewer 1 Report

Very comprehensive review in ED.

Table 1 is very good and is it possible to have another table to descript the 

alarming effects?

The content about the EDs’ Ingredients and sport performance can be more concise.

Author Response

Point 1: Very comprehensive review in ED. Table 1 is very good and is it possible to have another table to descript the alarming effects?

Thank you for your positive feedback. This is a very good point that was missing, so we have added the suggested table describing the adverse effects of energy drink consumption.

Point 2. The content about the EDs’ Ingredients and sport performance can be more concise.

Thank you very much for your suggestion. However, that section of EDs’ content was focused on caffeine and taurine, which are the most common ingredients in EDs. We tried to explain widely if the combination of those substances triggers any effect and whether their co-administration is reasonable. The description of the rest ingredients were concise, as they do not occur often, their efficacy is limited or their exact quantities are not disclosed by producers. According to the sport performance, we specifically wanted to show that many studies had varied protocols what may explain why the results of studies are inconsistent. We intended to discuss mentioned findings in a broader context and we pointed out it in the manuscript.

Reviewer 2 Report

The communication is of most interest and high quality. There are some minor editing errors:

-Line 52, replace "on" with "one"

-Line 148, before plays it should read "it"

-Line 254, replace "influenced" with "influence"

-Line 276, omit one "of"

-Line 489, remove "another"

Author Response

Point 1: The communication is of most interest and high quality. There are some minor editing errors:

-Line 52, replace "on" with "one"

-Line 148, before plays it should read "it"

-Line 254, replace "influenced" with "influence"

-Line 276, omit one "of"

-Line 489, remove "another"

Thank you for your positive feedback and bringing these mistakes to our attention. All editing errors that you pointed out were corrected.

Reviewer 3 Report

It is a comprehensive text, the article is quite well written and complete. but you should make clear the aim of the study in the main text. 

It would be to add a table on the health effects of energy drinks.
Furthermore, more information about the ED's content and their effects on sports could be included and also the reason behind these effects.

Author Response

Point 1: It is a comprehensive text, the article is quite well written and complete. but you should make clear the aim of the study in the main text. 

Thank you for your positive feedback. We appreciate your suggestion, therefore we added a short paragraph explaining our aim of the study.

Point 2: It would be to add a table on the health effects of energy drinks.

This is a very good point that was missing, so we have added the suggested table describing the adverse effects of energy drink consumption.

Point 3: Furthermore, more information about the ED's content and their effects on sports could be included and also the reason behind these effects.

Thank you for that relevant suggestion. However, we decided to focus on caffeine and taurine, which are the most common ingredients in EDs. We tried to explain widely if the combination of those substances triggers any effects and whether their co-administration is reasonable. We presented their efficacy in some sports disciplines and described their potential biochemical effect on human body, especially on muscular and endocrine system. The description of the rest ingredients were concise, as they do not occur often, their efficacy is limited or their exact quantities are not disclosed by producers. In our opinion, too expanded discussion about all ingredients would disrupt proportion between particular sections of the manuscript.